



**Ionospheric total electron content anomaly possibly associated with the April 4,**
**2010 Mw7.2 Mexico earthquake**
Jing Liu[1,2], Wenbin Wang[2], and Xuemin Zhang[1]
*[1]Institute of Earthquake Forecasting, China Earthquake Administration, Beijing, China*
*[2]High Altitude Observatory, National Center for Atmospheric Research, Boulder, Colorado, USA*
Corresponding author: Jing Liu, Institute of Earthquake Forecasting, China Earthquake Administration,
No. 63 Fuxing Road, Haidian district, Beijing 100036, China. (liujingeva@163.com)
**Abstract**
Identifying ionospheric disturbances potentially related to an earthquake is a challenging work.
Based on the ionospheric total electron content (TEC) data from the madrigal database at the Haystack
Observatory, Massachusetts Institute of Technology, a new decomposition and nonlinear fitting method
has been developed and applied in this work to extract the TEC disturbances that are potentially related
to the Mw7.2 Mexico earthquake occurred on April 4 2010. By analyzing the TEC data for a long
period of time (72 days) before and after the earthquake, we found that a unique TEC depletion
occurred in the region around the epicenter on March 25. No other significant ionospheric TEC
anomalies were identified in the 72-day period around the earthquake, except some TEC disturbances
that appeared to be related to the geomagnetic activity between April 1 and 6, 2010. We further
analyzed the TEC data from other magnetically quiet days, and no TEC anomaly like that occurred on





March 25 was detected. The TEC data calculated from a first principles model SD-WACCM-X were
also analyzed using the same method as that for the observational data. No TEC anomaly was found on
March 25 from the model outputs either. Thus the source of the TEC anomaly on March 25 is unlikely
from the lower atmosphere waves. In this study, we show the occurrence of TEC anomaly on March 25,
10 days before the Mw7.2 Mexico earthquake and this TEC anomaly may not be explained by lower
atmosphere or geomagnetic activity forcing.
**Key words**: GPS TEC, ionospheric TEC anomaly, Mw7.2 Mexico earthquake
**1. Introduction**

The abnormal ionospheric density variations before and/or after earthquakes have attracted much

attention from the geophysicists for many years (e.g., Pulinets & Boyarchuk, 2004a; Le et al., 2015).
However, identifying and determining ionospheric density disturbances that are associated with an
earthquake have been challenging so far. For instance, Heki (2011) reported the enhancement of
ionospheric total electron content (TEC) ~40 min before the 2011 Mw9.0 Tohoku-oki earthquake.
However, in subsequent studies, some scientists questioned the true cause of this pre-seismic abnormity
in their comments (Kamogawa and Kakinami, 2013; Utada and Shimizu, 2014; Masci et al., 2015).
Heki and Enomoto (2013, 2015) later applied several data analysis methods and more case studies to
demonstrate the correlation between pre-seismic TEC abnormity and the earthquake. Despite these
controversies,    several    data    analysis    methods    have    been    employed    to    explore    potential



seismo-ionospheric perturbations in previous studies. The running mean method is a common approach
in analyzing time series data. Liu et al. (2000) utilized the running median of the critical frequency of
the ionospheric $F_2$ layer ($f$o$F_2$) and the inter-quartile range (IQR) of $f$o$F_2$ as the upper and lower bounds
to extract seismo-ionospheric precursors that may be associated with M≥6.0 earthquakes around Taiwan
from 1994 to 1999. Pulinets et al. (2005) calculated the monthly mean ($M$) of ionospheric TEC, and used
$M \pm \sigma$ as the thresholds to find TEC disturbances before the Colima Mexico earthquake, where $\sigma$ is the
standard deviation. In order to obtain the location characteristics of the ionospheric perturbations
associated with earthquakes, spatial analysis methods have been applied in some researches. Liu et al.
(2011) studied the locations of extreme TEC anomalies (enhancements or depletions) in the 12 2-hour
intervals for a day. They compared the data with previous 30-day data in each grid to determine if they
are maximum or minimum values in that 30-day period, and to see whether these TEC anomalies occur
only nearby the earthquake region or randomly worldwide. Liu et al. (2016) found, by calculating the
spatial relative changes, that ionospheric TEC, electron and ion densities were simultaneously enhanced
at different altitudes near the epicenter of the 2005 Sumatra Indonesia Ms 7.2 earthquake. A correlation
analysis between different stations was applied to demonstrate the local disturbance near the epicenter.
Pulinets et al. (2004b) calculated the cross-correlation coefficient between two measurement points
located inside or outside the earthquake preparation zone, and found that the coefficient sharply dropped
before strong seismic shocks. Iwata and Umeno (2016) detected the preseismic TEC anomalies before





the main shock, foreshock and aftershock of 2011 Tohoku-Oki *Mw* 9.0 earthquake by correlation
analysis.

Statistics analysis of seismo-ionospheric disturbances has also been attempted by scientists when

there are sufficient data. Liu et al. (2010) utilized the *z*-test for 150 M≥5.0 earthquakes during
2001-2007 to try to correlate the change of the ionospheric equatorial ionization anomaly (EIA) with
earthquakes. Parrot (2012) applied a software to automatically detect the abrupt enhancement of ion
density observed by the Detection of Electro-Magnetic Emissions Transmitted from Earthquake
Regions (DEMETER) satellite. Based on the statistical analysis of 17,366 M>4.8 earthquakes, he found
that perturbations in ionospheric ion density before earthquakes are more obvious than those prior to
random selected pseudo-earthquake events. Therefore, there is not a unified and standard method to
extract ionospheric density anomalies that may be related to earthquakes.

The ionosphere shows strong variability of different temporal and spatial scales. This variability

can be of different sources, including the effects of large-scale lower atmospheric waves, geomagnetic
and solar activity, and possibly, the earthquakes. Some ionospheric oscillations have known periods and
zonal structures (e.g., Forbes et al., 2008; Pancheva & Mukhtarov, 2012; Luan et al., 2012). In this study,
we use a new approach to extract possible ionospheric anomalies related to earthquakes. We obtain
TEC residuals by removing the known and identified oscillations in the ionosphere TEC data. Since
earthquakes are mostly single occurrence events at particular locations and times, these TEC residuals





can manifest earthquake effects on the ionosphere better. We use the TEC data from the madrigal
database at the Massachusetts Institute of Technology (MIT) Haystack Observatory. The TEC data have
high temporal resolution from a significantly large number of Global Position System (GPS) stations
(about 1500 sites from 2000, now almost 6000 sites) all over the world. Therefore, the database can
provide high temporal and spatial resolution data for our analysis. In this paper, section 2 describes the
data and analysis method. In section 3, MIT TEC data before and after the Mw7.2 Mexico earthquake
occurred on April 4 2010 are analyzed to obtain ionospheric TEC perturbations. In section 4, we use
more observational data from other time periods and first principles numerical simulations by
SD-WACCM-X to show the uniqueness of the TEC disturbances that occurred before the Mw7.2
Mexico earthquake. Finally, the conclusions are drawn in section 5.
**2. Observations and the Method for Data Analysis**
Based on a network of worldwide GPS receivers, MIT TEC is calculated by using an automated
software suite (Rideout & Coster, 2006). It includes downloading data, determining satellite and
receiver biases, removing data outliers, mapping from slant TEC to vertical TEC, and so on. The data
are provided as estimates of vertical TEC in $1°$ by $1°$ grids distributed around locations where data are
available. The temporal resolution of the TEC maps is 5 minutes. The advantage of MIT TEC is that it
is strictly data driven with no underlying models that smooth out the real gradients in the TEC. In this





study, the TEC data are downloaded from the MIT Haystack Observatory madrigal database
(http://madrigal.haystack.mit.edu/madrigal/).

The Fast Fourier Transform (FFT) algorithm was applied to obtain the spectral distribution of the

TEC mean value in the northern American region (20°N-50°N in latitude, 90°W-140°W in longitude)
from 2000 to 2017 (Figure 1). Multi-day spectral peaks are seen in the figure, including 27-day solar
rotation, semiannual and annual oscillations. The high-frequency tidal spectral peaks at 24-hour
(diurnal), 12-hour (semidiurnal), 8-hour (terdiurnal) and 6-hour (quad diurnal) are also obvious in
Figure 1. The TEC data in each day can be expressed as a superposition of tide-like components (Forbes
et al., 2008; Luan et al., 2012). In this paper we used Eq. (1) to express TEC data, which includes 6
terms:

$$f(t)=A(0)+A(1)*\cos\left(\frac{2\pi}{24}t+B(1)\right)+A(2)*\cos\left(\frac{2\pi}{12}t+B(2)\right)+A(3)*\cos\left(\frac{2\pi}{8}t+B(3)\right)$$

$+A(4)*\cos\left(\frac{2\pi}{6}t+B(4)\right)+A(5)$                                     (1)
where A(0) is the daily mean TEC, A(1), A(2), A(3), A(4) and B(1), B(2), B(3), B(4) are the amplitudes
and phases of diurnal, semidiurnal, terdiurnal, and 6-hour oscillations, respectively. A(5) is the residual,
which includes higher frequency oscillations and/or some unknown variability, for example, the
perturbations caused by an earthquake. In this study, three steps were taken to obtain A(5). First, if there
were extremely large values in the raw data, which may be caused by data error, the data were canceled
at its observation time. Second, a linear fitting between the solar 10.7 cm radio flux index ($F_{10.7}$), the





geomagnetic activity index (*AE*) and the TEC data was applied to remove solar and magnetic activity
effects. Similar to Pi et al. (2003) and Lei et al (2004), who used nonlinear least square minimization to
minimize the difference between the model results and observational data to investigate the mechanisms
of ionospheric variations, we also employed a nonlinear fitting method to obtain the coefficients in Eq.
(1) for each day in the third step. The data would not be fitted if the number of data in a day is less than
72 (the total number of data is 288 for each day at the 5-minute cadence) and if the data gaps are larger
than 6 hours. A running mean method was applied to do these fitting, with 1-day window and 1-hour
step, which means there will be 24 fitted data in each day. From these three steps, A(5) was extracted
using Eq. (1) to analyze the TEC changes before and after the earthquake. Hereafter, we will show TEC
residual values of A(5) obtained from the above described data analysis processes.

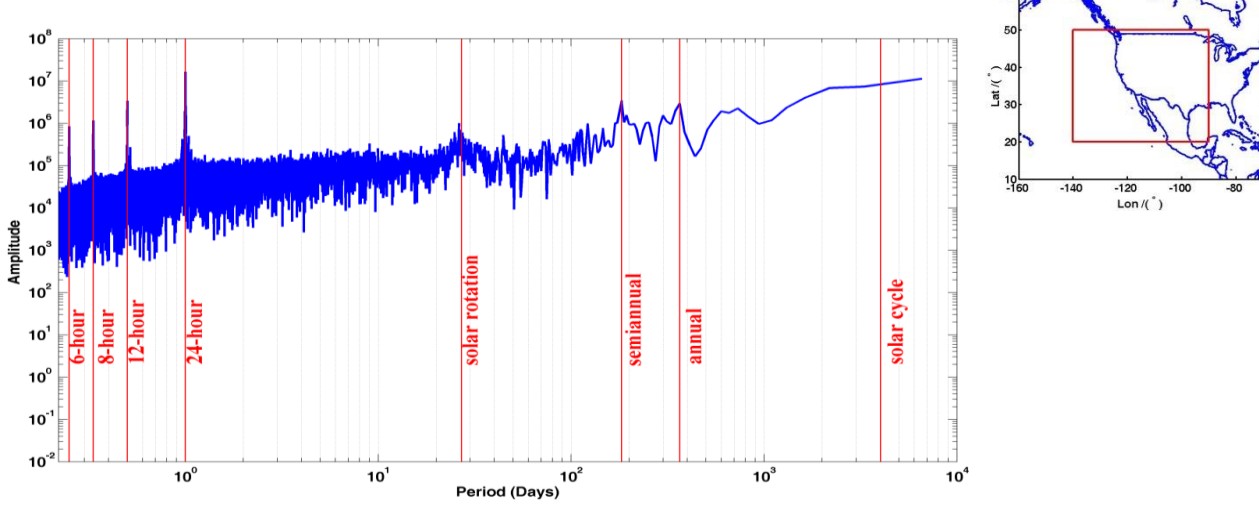


**Figure 1:** The FFT spectrum of the TEC mean values from 2000 to 2017 in the North American region (20°N-50°N in
latitude, 90°W-140°W in longitude), which is shown with the red rectangle in the subplot.





## 3. Results

    The Mexico Mw7.2 earthquake with 10 km depth occurred at 22:40 UT (universal time) on April 4

2010. The epicenter was located at (32.286°N, 115.295°W). The TEC data around the epicenter in a

region with latitude from 30°N-34°N and longitude from 113°W-117°W were obtained and analyzed

from March 14 to April 6, 2010. The mean TEC residual in this region is shown in the bottom panel of

Figure 2. From the time series of $F_{10.7}$ and geomagnetic activity indices ($K$p, $D$st, and the $AE$) in Figure

2, it can be seen that there was no geomagnetic activity from March 14 to March 31. It became more

active since April 1, especially after April 5 when $D$st dropped to below -40 nT.

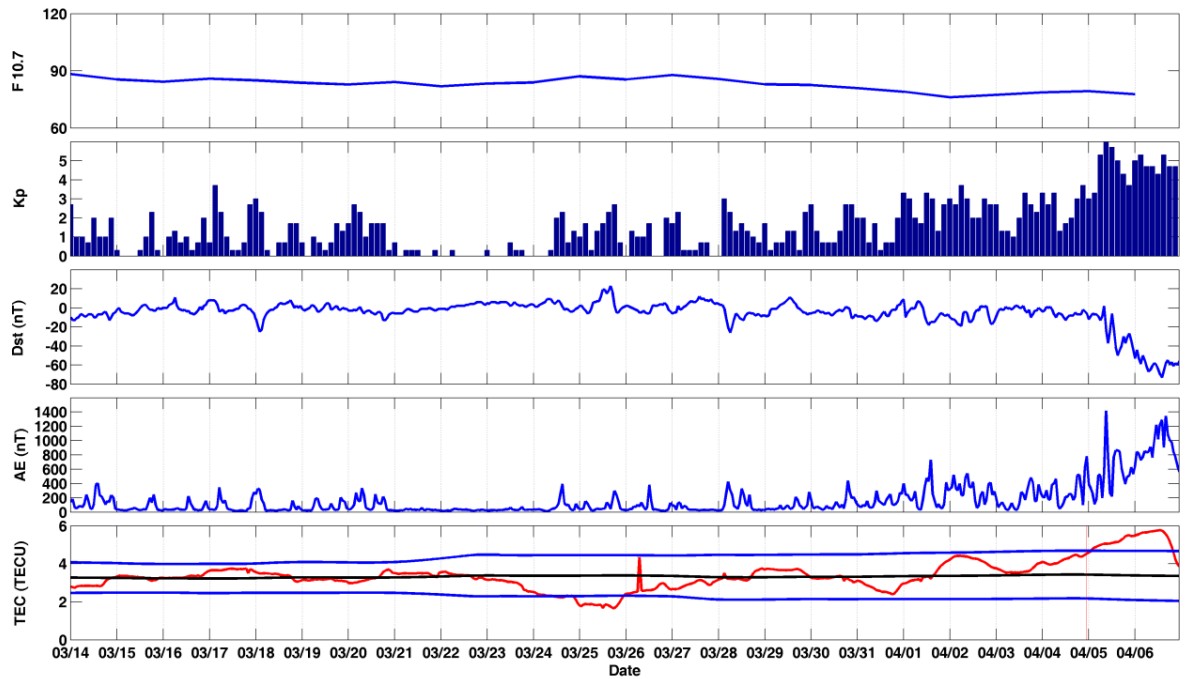

**Figure 2:** Time series of TEC residual (A(5)) around the epicenter from March 14 to April 6, 2010. Panels from top to

bottom represent time series of $F_{10.7}$, $K$p, $D$st, the $AE$ index and TEC residual, respectively. In the bottom panel, the red





line is the mean value of the TEC residual in the region of latitude 30°N-34°N and longitude 113°W-117°W. The black
and blue lines represent the mean values and M±1.5*$\sigma$ of ±15 days of data centered around a particular day. The
vertical red line on April 4 indicates the occurrence time of the Mexico Mw7.2 earthquake.
Under the assumption of a normal distribution, the probability of data in the range of ±$\sigma$ and ±2$\sigma$ is
68.26% and 95.44%, respectively. In order to avoid the probability being too low or high, we used
M±1.5*$\sigma$ (the probability is 86.64%) as the threshold to extract the disturbances that may be related to
earthquakes, where M and $\sigma$ stands for the mean value and standard deviation of TEC residuals of ±15
days centered around a particular day, respectively. Before the Mexico earthquake, the TEC value was
lower than the threshold on March 25. In other days from March 14 to April 5, TEC residual value for
each day was within the thresholds. When magnetic activity became stronger, the TEC was also
increasing with the values over the upper bound from April 5 to 6. Therefore, except the depletion of
TEC residual on March 25 and the increase of TEC residual associated with the magnetic activity on
April 5 and 6, no TEC anomalies exceeding the thresholds were detected in other days during the
72-day period.
In order to further analyze the TEC changes potentially related to the earthquake, we expanded the
region of interest to include the area of latitude 20°N-50°N and longitude 90°W-140°W. In this analysis
of TEC spatial structure, the differences between the TEC residuals and the mean values of ±15-day
data for a particular day were obtained before and after the earthquake. For each day, the mean value of





the 24-hour data was used to represent the TEC residual, as shown in Figure 3. The TEC depletion on
March 25 is evident in the region surrounding the epicenter, similar to the analysis result of the time
series shown in Figure 2. The TEC depletion can also be seen south and west to the epicenter. In all the
days shown in Figure 3, only on March 25 did the TEC residual data show depletion in the region
around the epicenter. The TEC residuals began to increase from April 1 in the western and southern part
of the region. Large TEC values occurred in a large region when magnetic activity became strong on
April 5 and 6.

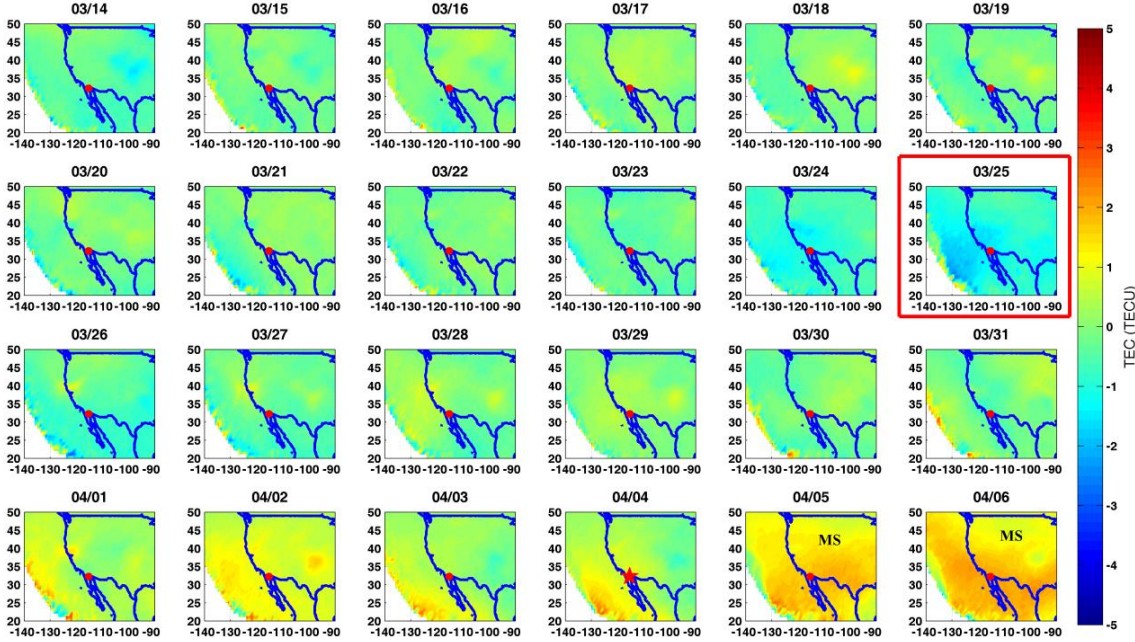


**Figure 3:** TEC changes in the region of latitude 20°N-50°N and longitude 90°W-140°W from March 14 to April 6,
2010. The red dot indicates the epicenter. The red star shows the day of the earthquake. The blue lines represent the
continent. 'MS' on April 5 and 6 mean 'magnetic storm'. The date of the map is marked on the top of each subpanel.
The subpanel on March 25 is highlighted with the red rectangle.





## 4. Discussion

In order to further establish the possible correlation between the TEC depletion on March 25 and
the Mexico Mw7.2 earthquake on April 4, we carried out a number of detailed analysis of the TEC
variations in the region. Firstly, the analysis time was extended to more days before March 14 and after
April 6 to determine the TEC changes over a longer period of time. The results are given in Figures 4
and 5. We can see that, except the TEC decrease on April 11 when a magnetic storm occurred with a
minimum $D$st value of -55 nT, there were no extremely high or low TEC values in the region for all the
days. Therefore, we can see from Figures 3-5 that, in 72 days from February 18 to April 30, there were
four days of TEC anomalies: TEC depletion on March 25 under geomagnetically quiet conditions, TEC
enhancements on April 5 and 6 under geomagnetically active conditions, and TEC depletion on April
11 under geomagnetic storm conditions. The unique occurrence of TEC depletion on the
geomagnetically quiet day of March 25 is thus potentially connected to the occurrence of the earthquake
on April 4 in the same region.



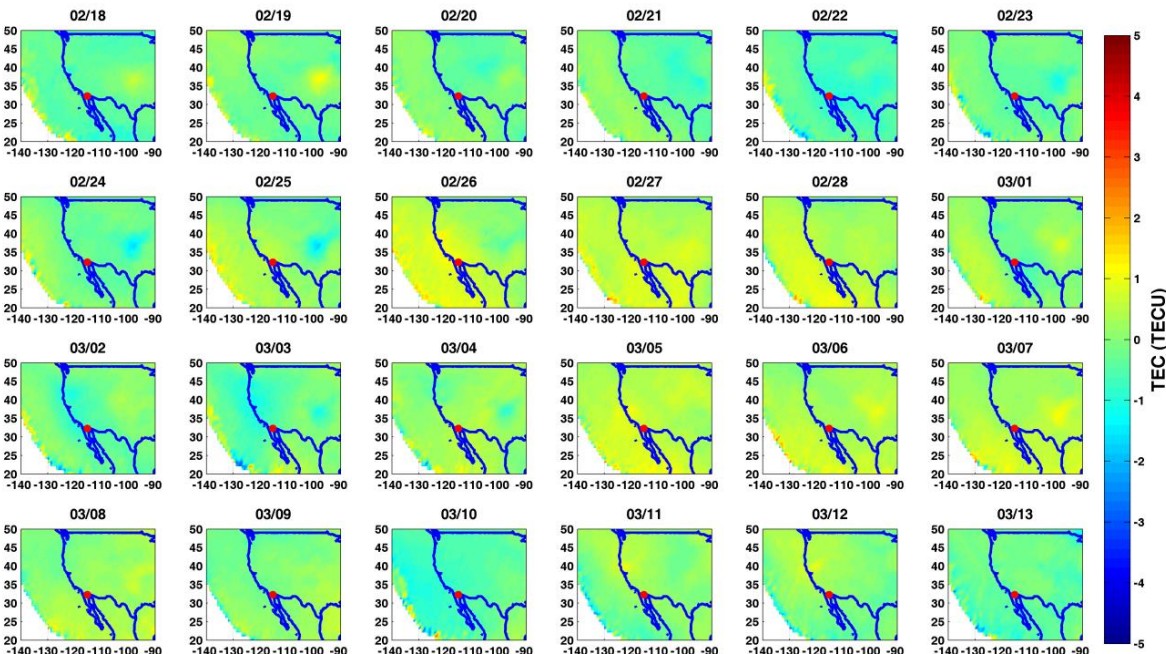


**Figure 4:** Same as Figure 3, but for February 18 to March 13, 2010.

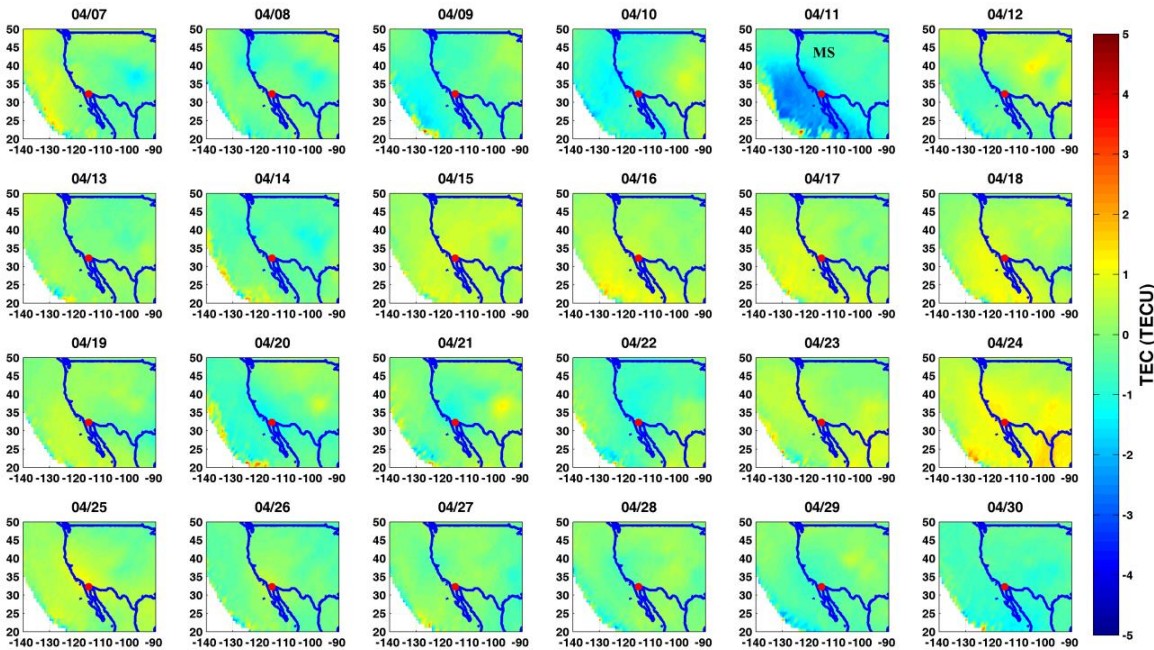


**Figure 5:** Same as Figure 3, but for April 7 to April 30, 2010.




Secondly, we analyzed the TEC changes in the geomagnetically quiet days of other years (-30 nT <
$D$st < 20 nT, $K$p < 3, $AE$<500 nT) using the same analysis method described above. We found that the
time period from November 27 2009 to January 19 2010 was geomagnetically quiet and there were
continuous TEC data to allow a meaningful data analysis. In this study, since we use ±15 days data as
the background, the first day with analysis results should be 15 days later than November 27 2009. The
distribution of the differences of TEC residuals from December 12 2009 to January 4 2010 is given in
Figure 6. There was no TEC anomaly in all those days, which means that when geomagnetic activity is
quiet, the TEC anomaly seen on March 25 before the Mexico Mw7.2 earthquake may not appear.

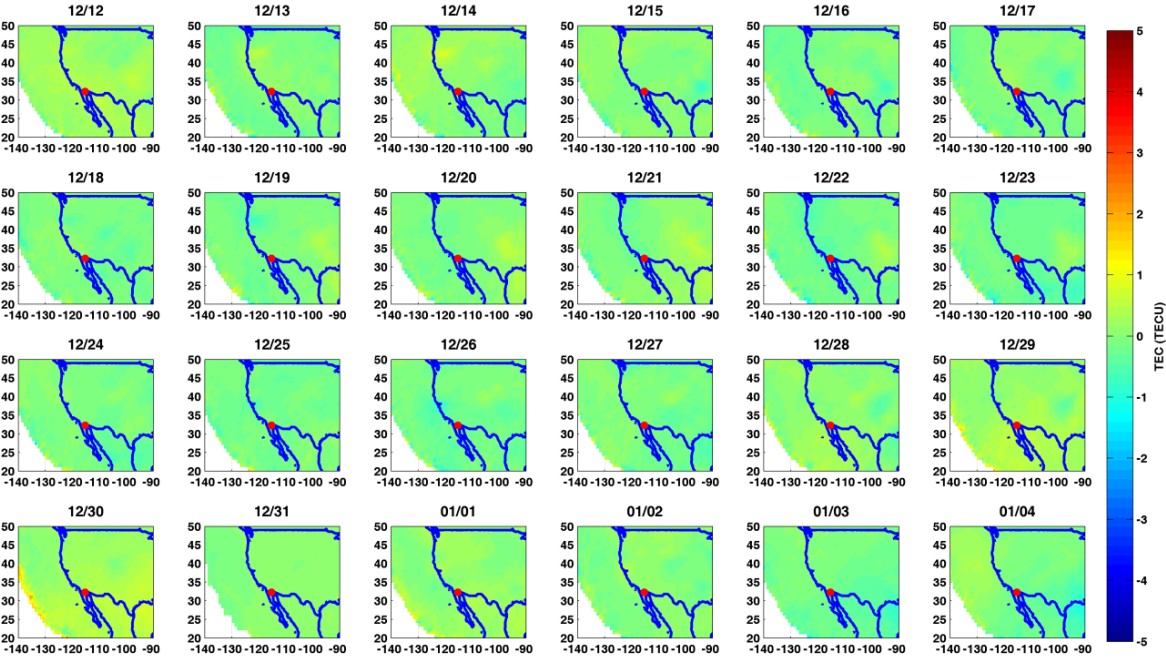


**Figure 6:** Same as Figure 3, but for December 12 2009 to January 4 2010 in geomagnetically quite days.





Thirdly, it is important to distinguish the seismic disturbance and the medium-scale traveling
ionospheric disturbances (MSTIDs). Iwata and Umeno (2017) calculated the rate of anomalous area and
the propagation velocity to detect the preseismic ionospheric disturbances. Otsuka et al. (2011) reported
that the propagation of atmospheric gravity waves and auroral activity are the main sources of the
MSTIDs. In this study, first principles simulations were employed to further examine the potential
source of the TEC anomaly seen on March 25, 2010, especially the lower atmospheric waves. Here we
used the thermosphere and ionosphere extension of the Whole Atmosphere Community Climate Model
(WACCM-X) (Liu et al., 2018). The top boundary of WACCM-X is set at $4.0 \times 10^{-10}$ hPa (~500 to ~700
km altitude, depending on solar activity). WACCM-X can well simulate the chemical and physical
processes in the atmosphere (Marsh et al., 2013; Neale et al., 2013). WACCM-X can be configured
either for free climate simulations (lower atmosphere unconstrained), or to have the tropospheric and
stratospheric dynamics constrained to meteorological reanalysis fields for specifically targeted time
periods. The later one is called specified dynamics WACCM-X (SD-WACCM-X, Sassi et al., 2013). A
detailed description of the SD-WACCM-X can be found in Marsh (2011). With the lower atmospheric
dynamics constrained by the observational data, SD-WACCM-X can accurately represent the
large-scale and medium-scale lower atmospheric waves that can propagate upward and affect the
thermosphere and ionosphere. In this study, we used the SD-WACCM-X to determine whether the TEC
depletion seen on March 25 is related to lower atmospheric forcing. The SD-WACCM-X has a





horizontal resolution of 1.9° in latitude and 2.5° in longitude. Using the same data analysis method
described in section 2, the distributions of the differences of the simulated TEC residual from March 14
to April 6, 2010 are shown in Figure 7 within the region of latitude 20°N-50°N and longitude
90°W-140°W to be consistent with the data. Except the TEC enhancements around the northern crest of
EIA on April 5, no TEC anomaly is identified around the epicenter. The TEC depletion on March 25 is
not detected in SD-WACCM-X outputs, which means that the TEC anomaly source may not result from
the lower atmospheric forcing.

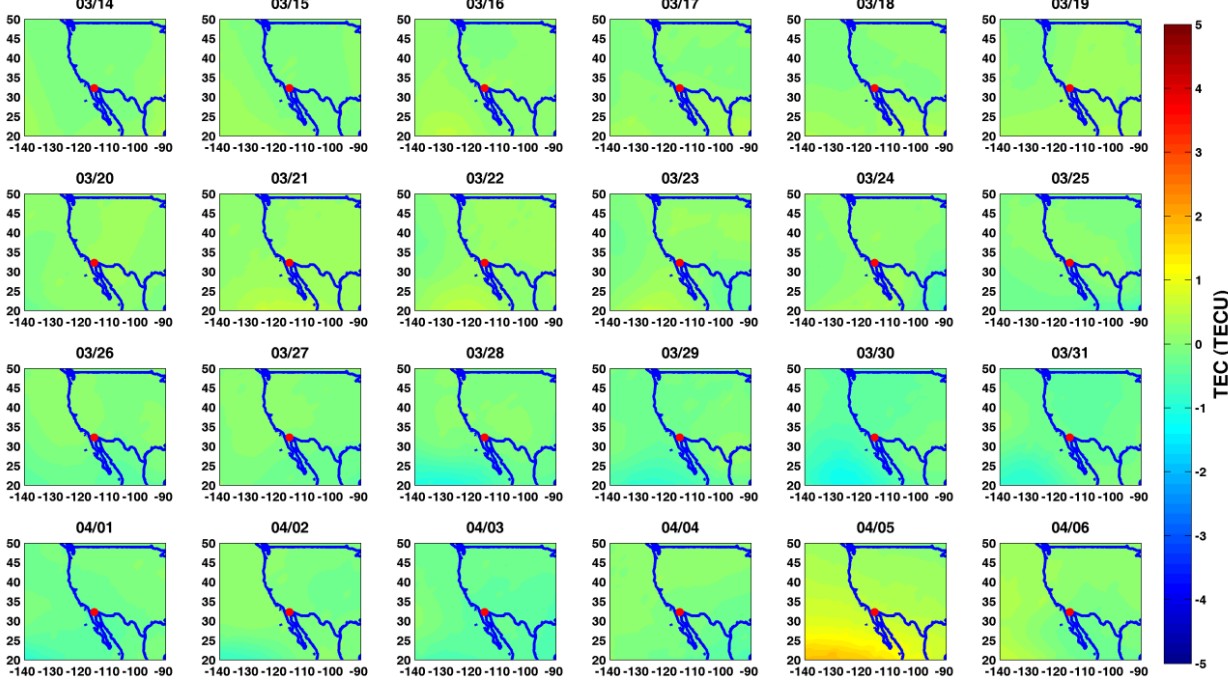


**Figure 7:** Same as Figure 3, but for SD-WACCM-X outputs from March 14 to April 6, 2010.





Takeuchi et al. (2006) pointed out that the stress of the p-holes in the crust can reach the surface
and create an upward electric field. Pulinets and Ouzounov (2011) supported the hypothesis of
atmospheric electricity changes caused by radon emanation and air ionization near the fault zone.
Sorokin et al. (2005) suggested that the DC electric field that forms above seismically active regions
could penetrate into the ionosphere. Kuo et al. (2011, 2014) proposed the electrical coupling between
the ionosphere and the surface charges in the earthquake fault zone. They suggested that the vertical
surface electric field drives currents in the atmosphere and electric fields at the bottom of the ionosphere.
If there is an upward electric field in the ionosphere, with the $\boldsymbol{E} \times \boldsymbol{B}$ drifts, the plasma moves westward.
Furthermore, Pulinets and Boyarchuk (2004a) suggested that the seismo-ionospheric phenomena did not
coincide with the vertical projection of the epicenter, but shifted equatorward. In our study, we found
that the depletion of TEC residuals occurred not only over the epicenter but also south and west to the
epicenter, which may be related to the above mentioned lithosphere-atmosphere-ionosphere coupling.
**5 Conclusions**
In this study, we applied a decomposition and nonlinear fitting method on the MIT TEC data to
obtain ionospheric TEC anomaly that is possibly associated with earthquakes. We analyzed the MIT
TEC data near the Mexico Mw7.2 earthquake that occurred on April 4 2010. We also carried out
numerical simulations using first principles model SD-WACCM-X. The main conclusions of this work
are as follows:



The TEC decreased on March 25 around the epicenter, 10 days before the earthquake. Except for
the TEC perturbations that were clearly related to geomagnetic activity, no TEC anomaly similar to that
on March 25 was seen in other 68 days around the day of the earthquake. Furthermore, the TEC
anomaly seen on March 25 cannot be found in geomagnetically quite days from December 12 2009 to
January 4 2010 in the same region, either.
The TEC simulated by the SD-WACCM-X runs did not show TEC decrease around the epicenter
on March 25. SD-WACCM-X includes lower atmospheric large-scale waves and their coupling effects
on the ionosphere. Therefore, the model results suggest that the ionosphere TEC anomaly on March 25
might not be the result of lower atmospheric forcing. Our data analysis and model simulations thus
indicate that the TEC anomaly seen on March 25 may be potentially related to the Mexico Mw7.2
earthquake.
Although we identify ionospheric anomalies (TEC depletion) that are possibly associated with the
2010 Mexico Mw7.2 earthquake in this work, more case studies and physics-based simulations are
needed in the future to fully understand the physical mechanism of the seismo-ionospheric coupling.
**Acknowledgments**
This work is supported by the National Key R&D Program of China (2018YFC1503506), the
APSCO Earthquake Research Project Phase Ⅱ, the China Scholarship Fund and ISSI-Beijing. The
authors acknowledge the madrigal database at the Massachusetts Institute of Technology Haystack



Observatory for providing the GPS TEC data, and the data can be downloaded from
http://madrigal.haystack.mit.edu/madrigal/. The authors acknowledge the WACCM-X development
teams at NCAR. The WACCM-X is an open-source community model. The outputs from model runs
used in this study are calculated using the NCAR supercomputer. NCAR is sponsored by the National
Science Foundation.
**Data availability**
The MIT TEC data can be downloaded from http://madrigal.haystack.mit.edu/madrigal/.
**Author contribution**
Jing Liu analyzed the data and writed the manuscript. Wenbin Wang proposed the topic, conceived
and designed the study. Xuemin Zhang helped in the interpretation. All authors read and approved the
final manuscript.
**Competing interests**
The authors declare that they have no competing interest.
**Statement**
The journal of Annales Geophysicae has the right to reproduce materials in this manuscript,
including figures, tables, and maps.

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
