# Peer review of "Ionospheric total electron content anomaly possibly associated with the April 4, 2010 Mw7.2 Mexico earthquake"

_Annales Geophysicae, 2020_

## Referee Comment (RC1) · Anonymous Referee #1 · 11 Apr 2020

The paper of Liu et al. creates very strange impression starting from the item selected and finishing by used methodology of data processing. So, let's start from the very beginning. 1. Why in year 2020 was selected earthquake which took place 10 years ago and which was studies by other scientists: Mustafa Ulukavak & Mualla Yalcinkaya (2017) Precursor analysis of ionospheric GPSTEC variations before the 2010 M7.2 Baja California earthquake, Geomatics, Natural Hazards and Risk, 8:2, 295-308, DOI: 10.1080/19475705.2016.1208684

Y. B. Yao, P. Chen, S. Zhang, J. J. Chen, F. Yan, and W. F. Peng, Analysis of pre-earthquake ionospheric anomalies before the global M = 7.0+ earthquakes in 2010

[Figure]

Nat. Hazards Earth Syst. Sci., 12, 575–585, 2012

Actually, the case studies can be accepted now if something exclusive was detected or some original technology was applied. So let us consider what kind technologies of data processing and methodology of precursor identification were applied. 2. The only unique in the paper is the use of MIT TEC maps. Authors consider these maps probably as advantage because of "The advantage of MIT TEC is that it is strictly data driven with no underlying models that smooth out the real gradients in the TEC" in addition the maps have the higher temporal (5 min) and spatial (1ïĆřx1ïĆř) resolution in comparison with GIM TEC maps (IONEX). And here immediately some comments appear. Use of such kind of maps is possible if you have the distance between GPS receivers of order 100 km or less between them, so for such areas as oceans or Africa for example, such maps are not applicable. The linear regression without models is possible only if you have uniform distribution of receivers, otherwise you should use some interpolation procedures as Kriging, for example. So, the advantage of MIT TEC maps seems questionable. 3. My most concern is the use by authors the 24-hours averaging. This procedure could compared with calculating the average temperature of patients through the whole hospital. Ionospheric anomalies before earthquakes are transient phenomena and don't last through the whole day, So the average daily TEC is senseless. Such procedure may be applied probably with long lasting increase of F10.7 index, ore strong geomagnetic storm lasting several days, but not for ionospheric precursor's detection. Instead of use the mentioned by authors high temporal resolution of MIT TEC maps, they average them. In conclusion, I consider the obtained results questionable with application of not adequate technology of the precursor's identification and I'm forced do not recommend this paper for publication.

---

## Author Comment (AC1) · 23 Apr 2020

1. The paper of Liu et al. creates very strange impression starting from the item selected and finishing by used methodology of data processing. So, let's start from the very beginning. Why in year 2020 was selected earthquake which took place 10 years ago and which was studies by other scientists: Mustafa Ulukavak Mualla Yalcinkaya (2017) Precursor analysis of ionospheric GPSTEC variations before the 2010 M7.2 Baja California earthquake, Geomatics, Natural Hazards and Risk, 8:2, 295-308, DOI: 10.1080/19475705.2016.1208684 Y. B. Yao, P. Chen, S. Zhang, J. J. Chen, F. Yan, and W. F. Peng, Analysis of preearthquake ionospheric anomalies before the global

[Figure]

M = 7.0+ earthquakes in 2010. Actually, the case studies can be accepted now if something exclusive was detected or some original technology was applied. So let us consider what kind technologies of data processing and methodology of precursor identification were applied.

A: There are two reasons that we selected 2010 Mw7.2 Mexico earthquake to carry out this study. Firstly, in Fig. 1, it shows that there are more MIT TEC data in the North America region that makes it possible to determine unambiguously the potential earth quake signal in TEC and its regional distribution. Secondly, the seismo-ionospheric disturbances are likely related to the depth and magnitude of the earthquake (Le et al., 2011; Liu et al., 2014), therefore, the shallowest ones with M≥7.0 (shown in Table of Fig. 2) are selected in most studies (from 2000 to 2017). The 2010 Mw7.2 Mexico earthquake is thus a very suitable one for our study.

For the same earthquake event, new datasets and new analysis methods can be employed to obtain new results or insight. We believe that our paper is totally different from the early paper of Ulukavak Yalcinkaya (2017) in dataset (although both used TEC), analysis method, and results. In fact, a thorough examination of an event using different datasets and methods produce a more complete description of the events and gain new physical insight. Taking 2011 Tohoku-Oki Mw 9.0 earthquake as an example, many researchers have studied the seismo-ionospheric anomalies since its occurrence (Heki, 2011; Iwata and Umeno, 2016; Oyama et al., 2019). Specifically, Ulukavak and Yalcinkaya (2017) applied the time series method to analyze the original data of GPS TEC, while in our study, we use a new decomposition and nonlinear fitting method to extract possible ionospheric anomalies related to earthquakes. We obtain TEC residuals by removing the known and identified oscillations in the ionosphere TEC data. Since earthquakes are mostly single occurrence events at particular locations and times, these TEC residuals can manifest earthquake effects in the ionosphere better. Therefore, our method is completely different from that in Ulukavak and Yalcinkaya. Furthermore, we used physics-based whole atmosphere model simulations to demonstrate that the anomaly seen in TEC data is unlikely originated from lower atmospheric wave perturbations, which is definitely new and not in their paper either. We will cite this paper in the revised text and describe the differences between their method/results and ours.

2. The only unique in the paper is the use of MIT TEC maps. Authors consider these maps probably as advantage because of "The advantage of MIT TEC is that it is strictly data driven with no underlying models that smooth out the real gradients in the TEC" in addition the maps have the higher temporal (5 min) and spatial (1ïC′ ËĞrx1ïC′ ËĞr) resolution in comparison with GIM TEC maps (IONEX). And here immediately some comments appear. Use of such kind of maps is possible if you have the distance between GPS receivers of order 100 km or less between them, so for such areas as oceans or Africa for example, such maps are not applicable. The linear regression without models is possible only if you have uniform distribution of receivers, otherwise you should use some interpolation procedures as Kriging, for example. So, the advantage of MIT TEC maps seems questionable.

A: The advantage of our research is not only the data source, but also the analysis method. The TEC residuals are applied to extract anomalies associated with earthquakes by using a new decomposition and nonlinear fitting method, which is described in detail in the manuscript.

It is true that there are almost no data in the oceans and Africa, as shown in the Fig.1. The vertical TEC data of the map are obtained from slant TEC data, hence the distance between two GPS receivers may be a little farther than 100 km. In the North America, the GPS stations are sufficiently dense to obtain high spatial resolution maps, which is also the main reason that we selected 2010 Mw7.2 Mexico earthquake to do this analysis. Furthermore, as shown in Fig. 3 this earthquake occurred on the land and we have sufficient data to carry out our analysis.

3. My most concern is the use by authors the 24-hours averaging. This procedure could

compared with calculating the average temperature of patients through the whole hospital. Ionospheric anomalies before earthquakes are transient phenomena and don't last through the whole day, So the average daily TEC is senseless. Such procedure may be applied probably with long lasting increase of F10.7 index, ore strong geomagnetic storm lasting several days, but not for ionospheric precursor's detection. Instead of use the mentioned by authors high temporal resolution of MIT TEC maps, they average them. In conclusion, I consider the obtained results questionable with application of not adequate technology of the precursor's identification and I'm forced do not recommend this paper for publication.

A: In this study, the TEC residuals are applied to extract anomalies possibly associated with the earthquake by using a new decomposition and nonlinear fitting method. The high temporal resolution data is useful for the fitting method and, in fact, is used in our study. In other words, we use both high temporal resolution data and daily average in our paper, as explained below. The more the data, the better the fitting results.

At the beginning, the time series of TEC residual (extracted by the analysis method), as exhibited in Fig. 2 of the manuscript, is not averaged. Under the quiet geomagnetic activity conditions, the TEC value exceeded the threshold just on March 25, and this anomaly lasted for almost the whole day. Liu et al. (2011) found that the anomalous enhancement before 2010 M7 Haiti earthquake was lasting for about 31 hours. Next, in order to see the distribution of the anomalies, the TEC map is analyzed using the mean value of the 24-hour data for each day. It is seen that the TEC depletion on March 25 is not just in the epicenter but also in the surrounding area (Fig. 3 of the manuscript). Then, by analyzing the data in a long period of time and SD-WACCM-X simulations, we conclude that the TEC anomaly on March 25 cannot be explained by lower atmosphere waves or geomagnetic activity forcing. Therefore, we suggest the unique TEC depletion on March 25 is potentially related to the Mw7.2 Mexico earthquake occurred on April 4, 2010. Therefore, we did consider the time variation of the TEC, not just daily mean. The daily mean used is purely for the illustration of spatial distribution and we cannot show

a large amount of data with 5-minute cadence in the paper for the whole period. We will make this point clear in the next revised text. We apply our analysis method to extract the TEC disturbances and demonstrate that the TEC anomaly is possibly related to the Mexico earthquake. Therefore, our analysis method is new and the results of our study are important for the seismo-ionospheric research.

References Heki, K. (2011), Ionospheric Electron Enhancement Preceding the 2011 Tohoku-Oki Earthquake. Geophysical Research Letters, 38(17).

Iwata, T., Umeno, K. (2016), Correlation analysis for preseismic total electron content anomalies around the 2011 Tohoku-Oki earthquake. Journal of Geophysical Research: Space Physics, 121(9), 8969-8984.

Le, H., Liu, J. Y., Liu, L. (2011), A statistical analysis of ionospheric anomalies before 736 M6.0+ earthquakes during 2002–2010. Journal of Geophysical Research, 116(A2), A02303.

Liu, J., Le, H., Chen, Y., Chen, C., Liu, L., Wan, W., Su, Y., Sun, Y., Lin, C., Chen, M. (2011), Observations and simulations of seismoionospheric GPS total electron content anomalies before the 12 January 2010 M7 Haiti earthquake. J. Geophys. Res, 116, A04302.

Liu, J., Huang, J., Zhang, X. (2014), Ionospheric perturbations in plasma parameters before global strong earthquakes. Advances in Space Research, 53(5), 776-787.

Oyama, K. I., Chen, C. H., Bankov, L., Minakshi, D., Ryu, K., Liu, J. Y., Liu, H. (2019), Precursor effect of March 11, 2011 off the coast of Tohoku earthquake on high and low latitude ionospheres and its possible disturbing mechanism. Advances in space research, 63(8), 2623-2637.

Ulukavak, M., Yalcinkaya, M. (2016), Precursor analysis of ionospheric GPS-TEC variations before the 2010 7.2 Baja California earthquake. Geomatics Natural Hazards Risk.

[Figure]

**Fig. 1.** The distribution map of MIT TEC data

The list of M≥7.0 earthquakes from 2000 to 2017

| Time | Latitude | Longitude | Depth (km) | Magnitude |
|---|---|---|---|---|
| 2014-04-18T14:27:24.920Z | 17.397 | -100.972 | 24 | 7.2 |
| 2012-11-07T16:35:46.930Z | 13.988 | -91.895 | 24 | 7.4 |
| 2012-08-27T04:37:19.430Z | 12.139 | -88.59 | 28 | 7.3 |
| 2012-04-12T07:15:48.500Z | 28.696 | -113.104 | 13 | 7.0 |
| 2012-03-20T18:02:47.440Z | 16.493 | -98.231 | 20 | 7.4 |
| 2010-04-04T22:40:42.360Z | 32.28617 | -115.295 | 9.987 | 7.2 |
| 2010-01-12T21:53:10.060Z | 18.443 | -72.571 | 13 | 7.0 |
| 2009-05-28T08:24:46.560Z | 16.731 | -86.217 | 19 | 7.3 |
| 2005-06-15T02:50:54.190Z | 41.292 | -125.953 | 16 | 7.2 |
| 2003-01-22T02:06:34.610Z | 18.77 | -104.104 | 24 | 7.6 |

**Fig. 2.** The list of M≥7.0 earthquakes from 2000 to 2017

**Fig. 3.** The location of 2010 Mw7.2 Mexico earthquake

---

## Referee Comment (RC2) · Angelo De Santis (Referee) · 5 May 2020

This paper studied TEC anomalies potentially related to the 2010 M7.2 Mexico Earthquake. The article is reasonably well written, with a few exceptions (please see below). The content appears interesting, although some points are not clear or not well clarified or developed. I agree with the other referee about the fact that normally the application of a technique to a single study only could not be sufficient. However, the procedure to analyse TEC data is clear and partly original: this is, in my opinion, the most interesting contribution of the work, differing from most of the literature on this topic. I also liked the use of an atmospheric model to see if the found TEC anomaly on 25 March

2010 could have been produced by atmospheric forcing, not related to the impending earthquake.

In the following, I list some points that are fundamental before any publication. My final advice is a very major revision.

Major points

1. In general. In this article, specific information about the Mexico EQ is missing (tectonics of the region, fault style, effects of EQs in terms of deaths, economic losses, references, etc.). Also some literature on possible precursors of this EQ is missing.

2. Lines 64-65. You missed our recent publication on Scientific Reports (De Santis et al. 2019; https://doi.org/10.1038/s41598-019-56599-1), that proposes a unified and possible standard method.

3. Line 93. Is it the spatial mean at each time within the considered area? It is not clear. Could you please clarify?

4. Figure 2 (line 129). There is a clear spike in TEC data on 03/26. Why? Did you remove it before the analysis? It seems not.

5. Line 137. The reason to use M+/-1.5 sigma is not convincing. I would prefer at least 2 sigma. By the way, why do not you use median and IQR (e.g. 1.5 IQR), because ionosphere is very irregular and it does not have a Gaussian distribution around a mean?

6. Line 181. Why do not you show an analogous figure as Figure 2 also for the other period analysed as confutation period, i.e. December 12 2009 to January 4 2010? By the way, this confutation period is very short. Why do not add at least another period, too, with same quiet magnetic conditions?

7. Since you analyse the data considering 15 days before and 15 days after the day of concern, for estimating mean and sigma of the anomalous day of 25 March, you also

include the day of the earthquake, where a possible co-seismic effect in TEC could have been produced. Have you considered this? Do you think it did not affect your results? By the way, have you looked at it to see if some effect is visible?

8. Finally a remark. You find a single anomaly occurring around 10 days before the Mw7.2 earthquake. Why excluding the possibility of some other anomaly even well before, for instance in February, i.e. a month not analysed in this work? According to Rikitake law (the precursor time scales with earthquake magnitude) we would expect several months before for a such an earthquake.

Minor points.

9. Line 42. Please insert "2003" before "Colima Mexico earthquake".

10. Line 57. Please change "Statistics" with "Statistical".

11. Line 60. Please insert ""an original" before "software".

12. Line 91. This is the portal. Which is the precise site? At which date did you download the data? Please indicate better punctual information. Thanks.

---

## Author Comment (AC2) · 22 May 2020

We greatly appreciated Dr. Angelo De Santis for providing the evaluation and valuable suggestions to our manuscript. Here are the point-to-point replies.

1. In general. In this article, specific information about the Mexico EQ is missing (tectonics of the region, fault style, effects of EQs in terms of deaths, economic losses, references, etc.). Also some literature on possible precursors of this EQ is missing.

A: We will add more information about this earthquake in the revised version. The detailed text is as follows: 'The Mexico Mw7.2 earthquake with 10 km depth occurred

at 22:40 UT (universal time) on April 4 2010. The epicenter was located at (32.286N, 115.295W). It is also called M7.2 Baja California earthquake (Yao et al., 2012; Jie and Guangmeng, 2013; Ulukavak and Yalcinkaya, 2017). The earthquake occurred on the northwest-trending strike-slip fault, which is along the principal plate boundary between the North American and Pacific plates, with a movement rate of 4.6 mm per year (Ulukavak and Yalcinkaya, 2017). Most of the damage caused by this earthquake occurred in the twin cities of Mexicali and Calexico on the Mexico – United States border. At least three people lost their lives and about 100 people were injured in this nature hazard (Hermes, 2010).'

2. Lines 64-65. You missed our recent publication on Scientific Reports (De Santis et al. 2019; https://doi.org/10.1038/s41598-019-56599-1), that proposes a unified and possible standard method.

A: We will cite this article in the revised text in the 'Introduction' section: 'Based on the electron density and magnetic data observed by the Swarm constellation satellites for 4.7 years, statistical studies of 1312 M5.5+ earthquakes were carried out by De Santis et al. (2019). They found that ionosphere anomalies appear from a few days up to 80 days before the earthquakes, and that the occurrence of ionospheric anomaly is related to the earthquake magnitude.'

3. Line 93. Is it the spatial mean at each time within the considered area? It is not clear. Could you please clarify?

A: Yes, it is the spatial mean value for each 5-minute interval in the northern American region (20N-50N in latitude, 90W-140W in longitude). We will clarify this in the revised text: 'The Fast Fourier Transform (FFT) algorithm was applied to obtain the spectral distribution of the TEC spatial mean value for each 5-minute interval in the northern American region (20N-50N in latitude, 90W-140W in longitude) from 2000 to 2017 (Figure 1)'.

4. Figure 2 (line 129). There is a clear spike in TEC data on 03/26. Why? Did you

remove it before the analysis? It seems not.

A: In Fig. 2, we show the TEC residual change around the epicenter (in the region of latitude 30N-34N and longitude 113W-117W). Thank you for your comment. We reprocessed our data and found that there was a small bug in the processing routine. We re-plotted the data in Fig. S1, and the trend of TEC residual is almost the same as Fig. 2 in the text, except the spike on 26 March in the previous figure. So our results are not changed by this correction. In the revised text, we will replace with the new picture.

5. Line 137. The reason to use M+/-1.5 sigma is not convincing. I would prefer at least 2 sigma. By the way, why do not you use median and IQR (e.g. 1.5 IQR), because ionosphere is very irregular and it does not have a Gaussian distribution around a mean?

A: Under the assumption of a normal distribution, the probability of data within the range of $\pm\sigma$ and $\pm2\sigma$ is 68.26% and 95.44%. In order to avoid the probability being too low or too high, we chose MïĆś1.5* $\sigma$ as the threshold to extract the disturbances that may be related to the earthquake and the probability is 86.64%. There are also some researchers using mean values to identify the ionospheric disturbances associated with earthquakes, such as Pulinets et al. (2005).

According to the reviewer's suggestion, we tried to use the median values and 1.5*IQR as the threshold to re-plot Fig. 2, showing in Fig. S2. The depletion of TEC residual on March 25 is also obvious, almost the same as that in Fig. S1. Therefore, we believe the choice of M+/-1.5 sigma not affecting our analysis results.

6. Line 181. Why do not you show an analogous figure as Figure 2 also for the other period analysed as confutation period, i.e. December 12 2009 to January 4 2010? By the way, this confutation period is very short. Why do not add at least another period, too, with same quiet magnetic conditions?

A: We also applied the running mean method to analyze the time series changes of TEC residual, showing in Fig. 2. The advantage of this method is that it can reveal the data trend during a continuous time period, with 1 hour time resolution as in Fig. 2. Figure S3 shows the result of the time series analysis during the geomagnetically quiet period from December 12 2009 to January 4 2010. The variation of the observational data is very small so the data spread is much narrower, within 1 TECU. Although there are some data exceeding the thresholds, the maximum relative change is just 6ïïjĚ, whereas the relative change of TEC residual on March 25 for the Mexico earthquake is more 20%. Therefore, considering also the results presented in Figure 6, we do not think that those data outliers are an indication of earthquake related. On the other hand, this also highlights the disadvantage of this method: it does not provide the spatial characters of the data. Therefore, we applied the spatial analysis method to investigate the TEC residual changings in the region, such as in Fig. 3. In all the days shown in Fig. 3, only on March 25 did the TEC residual data show a depletion in the region around the epicenter. In order to compare with the TEC residual character on March 25, the spatial analysis method was also used in other time periods, such as those in Figs. 4, 5, 6, and 7.

In this study, since we use $\pm 15$ days data as the background, the time period of the data must cover almost 2 months. Applying the criteria of geomagnetically quiet conditions (-30 nT < Dst < 20 nT, Kp < 3, AE<500 nT), we survey the magnetic activity data between 2000 and 2017 and found that the time period from November 27 2009 to January 19 2010 was the only time period that satisfied the geomagnetically quiet conditions. Therefore, we showed the analysis results of this period in Fig. 6. We wish to have more geomagnetically quiet periods that could help determine whether the phenomenon on March 25 can be observed in other geomagnetically quiet periods without earthquakes, but constantly changing geophysical conditions makes it very difficult to realize.

7. Since you analyse the data considering 15 days before and 15 days after the day of

concern, for estimating mean and sigma of the anomalous day of 25 March, you also include the day of the earthquake, where a possible co-seismic effect in TEC could have been produced. Have you considered this? Do you think it did not affect your results? By the way, have you looked at it to see if some effect is visible?

A: Thank you for your suggestion. We also considered this issue in our analysis, while we didn't remove the data on the day of the earthquake for the data continuity. In our analysis, the background is moving, hence, if there are some effects, the data before and after the time period should be affected. However, we just found the TEC depletion on March 25, and no anomaly was seen before and after March 25.

8. Finally a remark. You find a single anomaly occurring around 10 days before the Mw7.2 earthquake. Why excluding the possibility of some other anomaly even well before, for instance in February, i.e. a month not analysed in this work? According to Rikitake law (the precursor time scales with earthquake magnitude) we would expect several months before for a such an earthquake.

A: In Figure 3, a large TEC depletion on March 25 was detected. In order to determine whether similar TEC changes occurred in a longer time period, the data of 72 days were also analyzed centered around the earthquake date. Besides the disturbance on March 25, no other significant ionospheric TEC anomalies were identified in the 72-day period around the earthquake, except some TEC disturbances that appeared to be related to the geomagnetic activity. We wish that we could examine the TEC anomaly for a longer period of time, as suggested by the reviewer. However, as the ionosphere changes greatly with geophysical conditions, including season, solar 27-day rotation and geomagnetic activity, to name a few, it is very difficult to extend the period beyond what we showed in this paper.

Pulinets and Boyarchuk (2004) summarized that the plasma density disturbances that are possibly related to earthquakes occur from about several days to a few hours prior to the earthquakes. Therefore, the time range of seismo-ionospheric anomaly analysis
should be long enough to extract the possible anomaly. Our approach is similar to previous studies. Liu et al. (2004; 2009; 2010) analyzed GPS TEC data ±15 days of the earthquakes to detect the seismo-ionospheric disturbances. Li and Parrot (2013) also paid attention to ±15 days of the earthquakes to analyze the ion density observed by the DEMETER satellite. Liu et al. (2011) used GPS TEC data 30 days before and 4 days after the 12 January 2010 M7 Haiti earthquake to study the seismo-ionospheric anomalies. Iwata and Umeno (2016) analyzed GPS TEC data 40 days before the 2011 Tohoku-Oki earthquake to check the pre-seismic TEC anomalies. The time range of our analysis covered 72 days (45 days before and 26 days after), for almost two and half months. Besides that, the TEC changes in other geomagnetically quiet days are also analyzed, which includes another 24 days of data. Therefore, we consider that the time range of our data analysis is long enough, as allowed by the required geomagnetically quiet conditions, for the purpose of study seimo-ionospheric connections.

9. Line 42. Please insert "2003" before "Colima Mexico earthquake".

A: We will insert it in the revised text.

10. Line 57. Please change "Statistics" with "Statistical".

A: We will modify it in the revised text.

11. Line 60. Please insert ""an original" before "software".

A: We will modify it in the revised text.

12. Line 91. This is the portal. Which is the precise site? At which date did you download the data? Please indicate better punctual information. Thanks.

A: The website is http://millstonehill.haystack.mit.edu/. We used the Matlab Madrigal remote data access programs provided by the website to download the data. You can press the 'APIs' on the website to see the detailed tutorial, and several popular programming languages (Matlab, python, and IDL) are available. You can also contact brideout@mit.edu, if there are any questions about the appropriate use and download

of these data.

Reference

Hermes, O.: 2010 Baja California earthquake, Alphascript Publishing, 2010.

Iwata, T., and Umeno, K.: Correlation analysis for preseismic total electron content anomalies around the 2011 Tohoku-Oki earthquake, J. Geophys. Res.Space Physics, 121, 8969-8984, doi:10.1002/2016JA023036, 2016.

Jie, Y., Guangmeng, G.: Preliminary analysis of thermal anomalies before the 2010 Baja California M7.2 earthquake, Atmósfera , 26, 473-477, doi:10.1016/S0187-6236(13)71089, 2013.

Li, M., and M. Parrot: Statistical analysis of an ionospheric parameter as a base for earthquake prediction, J. Geophys. Res. Space Physics, 118, 3731-3739, doi:10.1002/jgra.50313, 2013.

Liu, J. Y., Chen, Y. I., Jhuang, H. K., Lin, Y. H.: Ionospheric foF2 and TEC anomalous days associated with M$\geq$5.0 earthquakes in Taiwan during 1997-1999, Terr., Atm. Ocean Sci, 15, 371-383, 2004.

Liu, J. Y., Chen, Y. I., Chen, C. H., Liu, C. Y., Chen, C. Y.: Nishihashi, M., Li, J. Z., Xia, Y. Q., Oyama, K. I., Hattori, K., Seismoionospheric GPS total electron content anomalies observed before the 12 May 2008 Mw7.9 Wenchuan earthquake, Journal of Geophysical Research Atmospheres, 114(114), 231-261, 2009.

Liu, J., Chen, Y., Chen, C., Hattori, K.: Temporal and spatial precursors in the ionospheric global positioning system (GPS) total electron content observed before the 26 December 2004 M9. 3 Sumatra-a-Andaman Earthquake, J. Geophys. Res, 115, A09312, 2010.

Liu, J., Le, H., Chen, Y., Chen, C., Liu, L., Wan, W., Su, Y., Sun, Y., Lin, C., Chen, M.: Observations and simulations of seismoionospheric GPS total electron content

anomalies before the 12 January 2010 M7 Haiti earthquake, J. Geophys. Res, 116, A04302, 2011.

Pulinets, S. and Boyarchuk, K.: Ionospheric precursors of earthquakes, Springer Berlin Heidelberg, New York, 2004.

Pulinets, S. A., Contreras, A. L., Bisiacchi-Giraldi, G., Ciraolo, L.: Total electron content variations in the ionosphere before the Colima, Mexico, earthquake of 21 January 2003, Geofisica Internacional-Mexico, 44, 369-377, 2005.

Santis, A. D., Marchetti, D., Pavón-Carrasco, F. J., Cianchini, G.and Haagmans, R.: Precursory worldwide signatures of earthquake occurrences on Swarm satellite data, Scientific Reports, 9(1), 20287, https://doi.org/10.1038/s41598-019-56599-1.

Ulukavak, M., Yalcinkaya, M.: Precursor analysis of ionospheric GPS-TEC variations before the 2010 7.2 Baja California earthquake, Geomatics Natural Hazards & Risk, 1-14, 2016.

Yao, Y. B, Chen, P., Zhang, S., Chen, J. J., Yan, F., Peng, W. F.: Analysis of pre-earthquake ionospheric anomalies before the global M=7.0+ earthquakes in 2010, Nat Hazards Earth Syst Sci., 12, 575-585, 2012.

Please also note the supplement to this comment:
https://www.ann-geophys-discuss.net/angeo-2020-5/angeo-2020-5-AC2-supplement.pdf
* * *
[Figure]

**Fig. 1.** Figure S1: Time series of TEC residual (A(5)) around the epicenter from March 14 to April 6, 2010.

[Figure]

**Fig. 2.** Figure S2: Time series of TEC residual (A(5)) around the epicenter from March 14 to April 6, 2010, using median values and 1.5*IQR as the threshold.

[Figure]

**Fig. 3.** Figure S3: Time series of TEC residual (A(5)) around the epicenter from December 12 2009 to January 4 2010.